# The hydraulic efficiency of single fractures:
# Correcting the cubic law parameterization for self-affine surface roughness and fracture closure

Maximilian O. Kottwitz[1], Anton A. Popov[1], Tobias S. Baumann[1], and Boris J. P. Kaus[1]

[1]Johannes Gutenberg University, Institute of Geosciences, Johann-Joachim-Becher-Weg 21, 55128 Mainz, Germany

**Correspondence:** Maxmilian O. Kottwitz (mkottwi@uni-mainz.de)

**Abstract.** Quantifying the hydraulic properties of single fractures is a fundamental requirement to understand fluid flow in fractured reservoirs. For an ideal planar fracture, the effective flow is proportional to the cube of the fracture aperture. In contrast, real fractures are rarely planar, and correcting the cubic law in terms of fracture roughness has therefore been a subject of numerous studies in the past. Several empirical relationships between hydraulic and mechanical aperture have been proposed, based up on statistical variations of the aperture field. However, often they exhibit non-unique solutions, attributed to the geometrical variety of naturally occurring fractures.

In this study, a non-dimensional fracture roughness quantification-scheme is acquired, opposing effective surface area against relative fracture closure. This is used to capture deviations from the cubic law as a function of quantified fracture roughness, here termed hydraulic efficiencies. For that, we combine existing methods to generate synthetic 3D fracture voxel models. Each fracture consists of two random, $25cm^2$ wide self-affine surfaces with prescribed roughness amplitude, scaling exponent, and correlation length, which are separated by varying distances to form fracture configurations that are broadly spread in the newly formed two-parameter space (mean apertures in submillimeter range).

First, we performed a percolation analysis on 600'000 synthetic fractures to narrow down the parameter space on which to conduct fluid flow simulations. This revealed that the fractional amount of contact and the percolation probability solely depends on the relative fracture closure.

Next, Stokes flow calculations are performed, using a 3D finite differences code on 6400 fracture models to compute directional permeabilities. The deviations from the cubic law prediction and their statistical variability for equal roughness configurations were quantified. The resulting 2D solution fields reveal decreasing cubic-law accordance's down to 1 % for extreme roughness configurations. We show that the non-uniqueness of the results significantly reduces if the correlation length of the aperture field is much smaller than the spatial extent of the fracture. An equation was provided that predicts the average behaviour of hydraulic efficiencies and respective fracture permeabilities as a function of their statistical properties. A model to capture fluctuations around that average behaviour with respect to their correlation lengths has been proposed. Numerical inaccuracies were quantified with a resolution test, revealing an error of 7 %.

By this, we propose a revised parameterization for the permeability of rough single fractures, which takes numerical inaccuracies of the flow calculations into account. We show that this approach is more accurate, compared to existing formulations. It

can be employed to estimate the permeability of fractures if a measure of fracture roughness is available, and it can readily be incorporated in discrete fracture network modeling approaches.

## 1 Introduction

The geometrical inhomogeneities of single fractures and their effect on fluid flow remains a crucial parameter for understanding the hydraulic properties of fractured reservoirs, such as crystalline or tight carbonate rocks with nearly impermeable matrices. Hence, it has wide-ranging industrial applicability in the fields of petroleum and gas production, geothermal energy recovery, $CO_2$ sequestration, nuclear waste disposal, and groundwater management. Fluid flow in fractured reservoirs is commonly modeled by the discrete-fracture-network (DFN) approach (Bogdanov et al., 2003; Klimczak et al., 2010; Leung et al., 2012; de Dreuzy et al., 2012), which relies on knowing the permeability of single fractures. The permeability of a single fracture is often approximated by the well known cubic law (Snow, 1969; Witherspoon et al., 1980), assuming that a fracture is composed of two parallel plates separated by a constant aperture. However, natural fracture walls show deviations from planarity, i.e., roughness, resulting in varying apertures within the fracture plane. On top of that, fluid-rock interactions like dissolution (Durham et al., 2001), erosion (Pyrak-Nolte and Nolte, 2016) and mineral growth (Kling et al., 2017) as well as the surrounding stress field (Zimmerman and Main, 2004; Azizmohammadi and Matthäi, 2017) further modify the geometry of a fracture, causing deviations of the parallel plate assumption.

Considerable effort has been made to study the effect of fracture surface roughness on flow and reactive transport behavior. Early attempts (Patir and Cheng, 1978; Brown, 1987; Zimmerman and Bodvarsson, 1996; Oron and Berkowitz, 1998) employed the 2D Reynolds equation, a simplification of the Navier-Stokes equations, which assumes that the cubic law holds locally with the aperture varying in the $x - y$ along-fracture plane. They derived semi-empirical functions that describe the deviations from the cubic law in terms of the mean and standard deviation of the aperture field. Increasing computational power led to numerical improvements, with 3D Lattice Boltzmann (Jin et al., 2017; Foroughi et al., 2018) or Navier-Stokes (Mourzenko et al., 1995; Brush and Thomson, 2003) simulations revealing the non-uniqueness of previous functional approximations of fracture permeability. Factors such as shear displacement (Kluge et al., 2017), tortuosity, and the degree of mismatch between the opposing fracture surfaces (Mourzenko et al., 2018) were demonstrated to affect fluid flow paths and permeabilities. Detailed analyses of exposed fracture surfaces have shown that self-affine fractal models are capable of quantifying surface roughness properties from thin-section- to outcrop-scale. Thereby, the dependence of surface roughness as a function of observation scale is captured by their scaling exponent (the so-called Hurst exponent $H$). For example, mode I fractures in a porous sandstone showed $H$ of $0.4 - 0.5$ (Boffa et al., 1998; Ponson et al., 2007). Micro-fractures in Pomeranian shale featured $H$ of $0.3$ and $0.5$ (Pluymakers et al., 2017), depending on the opening mode. In other studies, $H$ of fault surfaces (mode II fractures) tends to fall in the range of $0.6 - 0.8$ with respect to slip orientation (Power and Tullis, 1991; Schmittbuhl et al., 1995; Bouchaud, 1997; Renard et al., 2006; Candela et al., 2012), regardless of rock type. Based on this, it is commonly assumed that a fracture consists of two opposing self-affine surfaces, and the resulting aperture field follows the same scaling relationship, assuming both surfaces are uncorrelated (Plouraboué et al., 1995). However, observations of opposing fracture

walls (Brown, 1995) have demonstrated that the two surfaces tend to be well correlated above a specific length scale and non-correlated below it, which poses an upper limit to the self-affine scaling in nature. Following Méheust and Schmittbuhl (2001, 2003), the ratio between system size $L$, and the correlation length $l_c$ defines whether the fracture has an intrinsic permeability or not. Their statistical approach suggested that permeabilities of uncorrelated fractures (i.e., $l_c/L = 1$) are strongly fluctuating and anisotropic for the same roughness configurations, revealing the importance of considering low $l_c/L$ ratios to be able to quantify an intrinsic fracture permeability.

Although extensively studied, no clear mathematical relationship between fracture roughness and permeability has been derived so far, leaving the cubic law as standardized parameterization in DFN modeling approaches. Thus, an applicable refinement is desired to promote their realism to help better understanding fluid flow on a reservoir scale. In this paper, existing algorithms are used to generate a large dataset of synthetic fractures covering all possible kinds of roughness configurations. Single-phase 3D Stokes flow calculations are then performed with a finite difference code, utilizing a high-performance-computing (HPC) cluster to handle the associated computational effort. By interpreting the statistical variations of the results, a refined parameterization of single fracture permeability is proposed, which is demonstrated to provide accurate predictions for the permeability of rough fractures.

## 2 Method & Data

### 2.1 Fluid flow in self-affine fractures

Generally, the flow of an incompressible Newtonian fluid is most accurately described by the Navier-Stokes equations (NSE). Assuming, that the flow is solely laminar (Reynolds numbers below unity according to Zimmerman and Bodvarsson, 1996), the fluid viscosity is constant and gravity is negligible at the system size, they reduce to the simpler Stokes equations, i.e., momentum balance (1) and continuity (2) equations, which for steady-state flow conditions are given in compact form by:

$$\mu \nabla^2 v = \nabla P, \tag{1}$$

$$\nabla \cdot v = 0, \tag{2}$$

with the fluid's dynamic viscosity $\mu$, pressure $P$ and velocity vector $v = (v_x, v_y, v_z)$, $\nabla$, $\nabla \cdot$, and $\nabla^2$ denote the gradient, divergence, and Laplace operator for 3D Cartesian coordinates, respectively.

These equations can be solved analytically for an idealized fracture, consisting of two parallel plates, vertically separated by a constant aperture $a$. Volumetric integration of the horizontally extended Poiseuille-flow solution yields the well known cubic law:

$$Q = -\frac{w a^3 \Delta P}{12 \mu}, \tag{3}$$

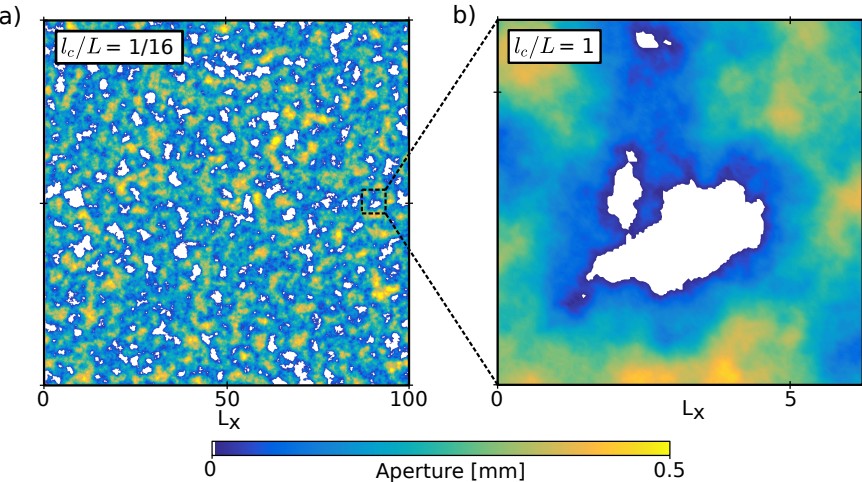

**Figure 1.** a) Aperture field of a synthetically generated fracture ($d = 1\ mm$, $\sigma_s = 0.5\ mm$, $H = 0.9$) with a $l_c/L$ ratio of $1/16$. b) Zoom-in showing the uncorrelated part of the aperture field. Units of $L_x$ are in $mm$.

with total volumetric flow rate $Q$, fracture width $w$ and pressure gradient along the fracture $\Delta P$ (see Zimmerman and Bodvarsson (1996) for a more detailed derivation). Combining equation 3 with Darcy's law for flow through porous media:

$$Q = -\frac{kA\Delta P}{\mu}, \tag{4}$$

with cross-sectional area $A$, leaves the intrinsic permeability $k$ of an idealized fracture proportional to its aperture by $k \propto a^2/12$. For a rough walled fracture, the aperture is no longer constant but rather varying across the fracture plane. The mean planes of an upper and lower rough surface $s_u(x,y)$ and $s_l(x,y)$ are separated by a constant distance $d$ to form $a(x,y)$ according to:

$$a(x,y) = \left[s_u(x,y) + \frac{d}{2}\right] - \left[s_l(x,y) - \frac{d}{2}\right]. \tag{5}$$

Depending on $d$, the surfaces may overlap at certain points and form contact areas, such that $a_0(x,y)$ is zero at these locations:

$$a_0(x,y) = \begin{cases} a & \text{if } a(x,y) > 0 \\ 0 & \text{if } a(x,y) \leq 0. \end{cases} \tag{6}$$

Assuming, that $s_u$ and $s_l$ are self-affine, the standard deviation $\sigma$ of the aperture field computed at system size $l$ scales as (Brown, 1995; Schmittbuhl et al., 1995):

$$\sigma = \begin{cases} \beta l^H & \text{if } 0 \leq l \leq l_c \\ \beta l_c^H & \text{if } l_c \leq l \leq L, \end{cases} \tag{7}$$

with the maximal system size $L$ and the correlation length $l_c$. Below $l_c$, the fracture is uncorrelated, and it is well correlated above it (see Fig. 1 for a visual explanation). The prefactor $\beta$ delivers information about the overall amplitude of the surface

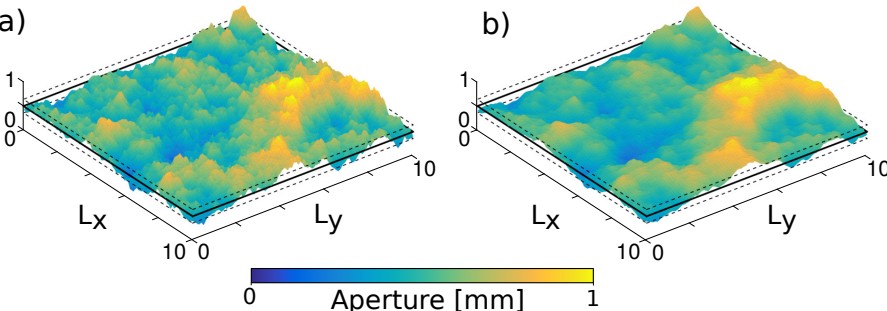

**Figure 2.** Two aperture fields constructed from synthetic fractures. Both aperture fields are based on the same sets of random numbers with varying Hurst exponents $H$, which is a) 0.4 and b) 0.8. The two statistical parameters $\bar{a}$ and $\sigma_a$ are indicated by black solid and dashed lines, respectively. Axis units are in $mm$, while the vertical axis (indicating aperture) is exaggerated by a factor of two for clarity. Note that $\bar{a}$ and $\sigma_a$ are identical for a) and b). Increasing height fluctuations at smaller scales, caused by a lower Hurst exponent results in a larger effective surface area $S$ for fracture a) compared to b).

roughness. $H$ typically denotes the scaling or Hurst exponent with $0 < H \leq 1$, whereas $H = 1$ corresponds to self-similar and $H < 1$ to self-affine scaling (e.g., Mandelbrot , 1982). Physically, self-affinity is expressed by higher height fluctuations at

smaller scales, leaving $H$ as a measure for the ratio of large scale versus small scale roughness intensity.

Here, we use the following two non-dimensional quantities to quantify the geometry of a rough fracture: (i) the relative closure $R$ and (ii) the effective surface area $S$. We compute the relative closure by dividing the standard deviation of the aperture field at the maximal system size $\sigma_a$ by the average aperture field $\bar{a}$:

$$R = \frac{\sigma_a}{\bar{a}}, \tag{8}$$

with $\bar{a}$ defined by:

$$\bar{a} = \frac{1}{L_x L_y} \int\limits_{x=0}^{L_x} \int\limits_{y=0}^{L_y} a_0(x,y)\,dx\,dy. \tag{9}$$

This quantity or its reciprocal is commonly used to infer the amount of contact between the opposing fracture walls (Patir and Cheng, 1978; Brown, 1987; Zimmerman and Bodvarsson, 1996; Méheust and Schmittbuhl, 2000). Theoretically, it falls in the

range $0 < R \leq \infty$, whereas $R = 0$ shows perfect accordance with parallel plates and the surfaces are in contact if $R \geq (3\sqrt{2})^{-1}$ (see Brown, 1987).

Furthermore, one requires a parameter that quantifies the effective surface roughness of a fracture since fractures with different $H$ can have equal $R$ values (see Fig. 2 for a visualization of the non-uniqueness of $R$). We, therefore, introduce a new quantity, the "effective surface area $S$". This parameter uniquely combines varying amplitudes and scaling exponents, because an in-

crease in fracture surface area is the direct consequence of increasing roughness. For that, we calculate the ratio of the surface

area of the fracture $sa_f$ to twice the area of its projection on the fracture plane (i.e., two times the base area perpendicular to the flow direction) $sa_c$ and normalize it with the fractional amount of the aperture field that has opened, i.e.:

$$S = \left(\frac{sa_f}{sa_c}\right)\left(\frac{1}{1-c}\right),$$ (10)

with $c$ being the contact fraction of the aperture field (i.e. where $a_0(x,y) = 0$), which leaves $1 \leq S \leq \infty$, with $S = 1$, showing perfect accordance with parallel plates. To finally quantify the influence of fracture roughness on its intrinsic permeability, the proportionality resulting from the cubic law needs to be corrected (e.g., Witherspoon et al., 1980) by applying a correction factor according to:

$$k \propto \chi \frac{a^2}{12}.$$ (11)

The approximation of $\chi$ in terms of quantified fracture roughness, i.e. $\chi(R,S)$, is the main subject of this study.

## 2.2 Numerical permeability estimation

To simulate the laminar flow of an incompressible, isothermal, and isoviscous fluid, we use a 3D binary voxel model input. We solve the linearized momentum balance (eq. 1) and continuity (eq. 2) equation in 3D, using velocity and pressure as primary variables. The coupled system is implemented in the open-source software package LaMEM (Kaus et al., 2016). LaMEM is a 3D staggered grid finite-difference code, which is based upon PETSc (Balay et al., 2018). The software is massively parallel and thus optimized for the use of high-performance computing (HPC) clusters to enable the computation of high-resolution models in a reasonable amount of time. Here, the matrix is considered impermeable and constrained to no-flow conditions by forcing all three velocity components to be zero at the matrix-void-boundary. Besides, the staggered grid discretization scheme

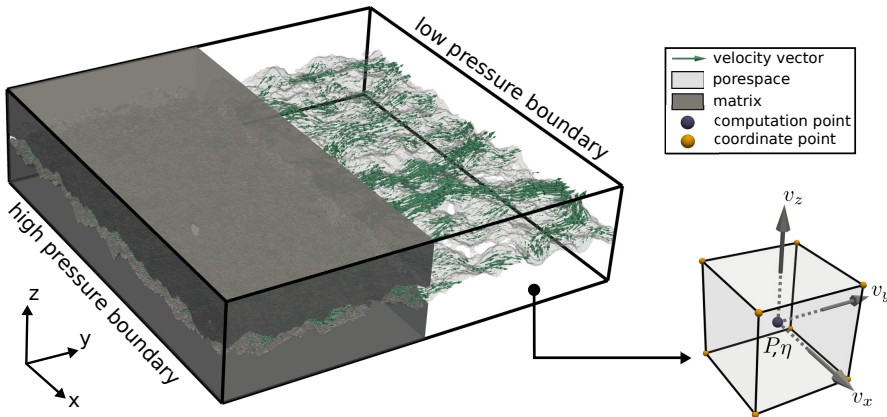

**Figure 3.** Model setup employed in the numerical simulations. The two boundaries where the pressure gradient is applied are indicated. The green velocity vectors are used for the computation of $\bar{v}$ and scaled according to their magnitude. The subfigure illustrates the staggered-grid discretization-scheme of a single voxel.

is rescaled at the fluid-matrix interface to provide higher accuracy (Eichheimer et al., 2019). Different pressures are applied on two opposing boundaries ($\Delta P = 0.01$ Pa for all models), while the remaining boundaries are set to no-slip. This fixes the

principal direction of fluid movement (here it is in y-direction, e.g. Fig. 3). After ensuring that the numerically converged solution is obtained (see appendix A in Eichheimer et al., 2019), the velocity component parallel to the principal flow direction is integrated over the volume to compute the volume average velocity $\bar{v}$ according to:

$$\bar{v} = \frac{1}{V} \int\limits_{V} |v_y| \, dy, \tag{12}$$

with the domain volume $V$. To finally obtain the intrinsic permeability $k_i$, $\bar{v}$ is substituted into Darcy's law for flow through
porous media, similar to the approach of Osorno et al. (2015):

$$k_i = \frac{\mu \bar{v}}{\Delta P}, \tag{13}$$

with the fluid's dynamic viscosity $\mu$.

## 2.3 Synthetic fracture dataset

As in-situ data of fractures are rarely accessible and limited to the size of drill cores, numerical studies commonly rely on a
stochastic generation approach for synthetic fractures (e.g., Brown, 1995; Candela et al., 2010). Here, we numerically generate isotropic self-affine surfaces with a MATLAB script (Kanafi, 2016), which makes use of random theory and fractal modeling techniques (Persson et al., 2004). It uses the standard deviation of surface heights $\sigma_s$, the Hurst exponent $H$, the cutoff length $l_c$, and the system size $L$ in $x$ and $y$ direction as input parameters to obtain $s_u(x,y)$ and $s_l(x,y)$, which are built from two independent Gaussian random number fields. The code is slightly modified, such that the seeds for the random number gen-
erator are prescribed to produce reproducible results. The mean planes ($x - y$ coordinate plane in both cases) of $s_u$ and $s_l$ are separated by varying values of $d$ according to equation 5 to simulate different closure stages of the fracture. Since ignoring mechanical deformation is a common practice (Brown, 1987; Méheust and Schmittbuhl, 2000, 2003; Mourzenko et al., 2018), we

**Table 1.** Minimal and maximal input values for parameters $d$, $\sigma_s$, $H$ and $l_c/L$. $n$ denotes the total number of increments, including minimum and maximum. Subscripts $g1$ and $g2$ indicate data for group 1 or 2, respectively. Thus, multiplying the $n$ values of each parameter gives the total number of parameter combinations (6000 for group 1, 320 for group 2). The number of realizations for a set (i.e. the number of different random number seeds used to generate the surfaces with one peculiar parameter combination) are given in the footnotes, resulting in a total of 600000 and 6400 fracture configurations for group 1 and 2, respectively.

| Parameter | Dimension | $\min_{g1}$ | $\max_{g1}$ | $n_{g1}$ | $\min_{g2}$ | $\max_{g2}$ | $n_{g2}$ |
|---|---|---|---|---|---|---|---|
| $d$ | mm | 0.01 | 1 | 20 | 0.2 | 5 | 4 |
| $\sigma_s$ | mm | 0.1 | 0.6 | 6 | 0.2 | 0.5 | 4 |
| $H$ | - | 0.1 | 1.0 | 10 | 0.1 | 1.0 | 4 |
| $l_c/L$ | - | 1/16 | 1 | 5 | 1/16 | 1 | 5 |

$r_{g1} = 100$, $r_{g2} = 20$

also assume that both surfaces are in contact at the locations where they overlap. Finally, the data is transferred into a 3D voxel space of $512 \times 512 \times 128$ voxels with a fixed physical voxel size of $0.1 \ mm$, resulting in a model domain of $51.2 \times 51.2 \times 12.8$

$mm$. The relative closure and the effective surface area are computed according to equations 8 and 10, respectively. It is important to note that both quantities are computed only within the effective pore space of the fracture parallel to the direction of the applied pressure gradient because only this contributes to the overall flow. For some configurations, it might be possible to have small amounts of trapped pore space within the fracture, that must be excluded before further numerical treatment. We separate the datasets generated in this study into two groups, each wich specific sets of input-parameter combinations and

certain number of realizations (input values are listed in table 1). The first group is used to analyze the percolation probabilities and determining the parameter boundaries for geometries of group 2, which are later used for for numerical flow simulations. To check, whether each fracture configuration in group 1 is able to transmit fluids, i.e., if it is percolating or not, we apply a recursive flood-filling MATLAB routine (e.g., Torbert, 2016). Then, the percolation probability $p$ represents the mean value of $n$ fracture realizations built from one specific input parameter combination, such that:

$$p = \frac{1}{n}\sum_{i=1}^{n} p_i \ , \ \text{with} \ p_i = \begin{cases} 1 \ \text{if percolating} \\ 0 \ \text{if non-percolating.} \end{cases} \tag{14}$$

Figure 4 shows the percolation probability as a function of relative closure $R$, as there was no notable variation with respect to their effective surface areas. Generally, the percolation probability starts reducing from 1 at $R \approx 1$ and converges to zero at $R \approx 5$. Higher $l_c/L$ ratios show earlier convergences to their percolation limit, as visible from the two fitted lines for $l_c/L$ 1

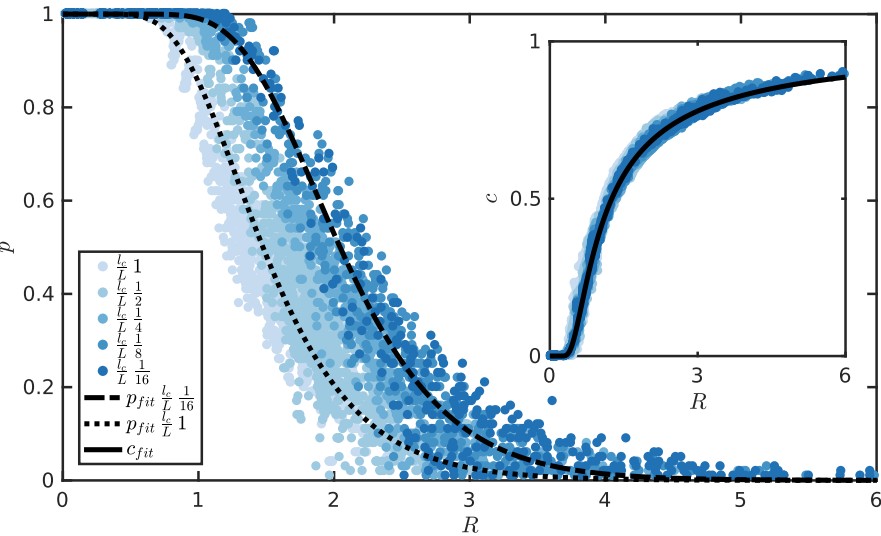

**Figure 4.** Percolation probability $p$ (eq. 14) and the mean fractional amount of contact $c$ as a function of relative closure $R$ for all fracture realization sets of group 1 and 2. Different shades of blue indicate different $l_c/L$ ratios as given in the legend, whereas black lines show best fits to the data.

**Table 2.** Resulting minimal and maximal values for mean aperture ($\bar{a}$), standard deviation of the aperture field ($\sigma_a$), contact fraction ($c$), relative fracture closure ($R$), effective surface area ($S$) and numerical fracture permeability ($k_m$) for the fracture geometries in group 2.

| Parameter | $\bar{a}$ | $\sigma_a$ | $c$ | $R$ | $S$ | $k_m$ |
|---|---|---|---|---|---|---|
| Dimension | $m$ | $m$ | - | - | - | $m^2$ |
| min | $1.91 \times 10^{-4}$ | $1.16 \times 10^{-4}$ | 0 | 0.03 | 1.04 | $6.78 \times 10^{-14}$ |
| max | $4.96 \times 10^{-3}$ | $8.51 \times 10^{-4}$ | 0.44 | 0.99 | 2.49 | $7.94 \times 10^{-7}$ |

and $1/16$. From the inset plot, it is evident that the contact fraction of all models only depends on the relative closure $R$, first contact between both walls occurs at $R \geq 3\sqrt{2}\sigma_a$, which is in good accordance with Brown (1987). Following this, we have chosen to limit the fracture geometries for the fluid flow simulations to configurations with $R \leq 1$ to (i) exclude non-percolation systems and (ii) limit the effect of the above-mentioned "melting" hypothesis, which intensifies with increasing R. To ensure applicability to nature, the input values for group 2 (see table 1) are chosen, such that the resulting fracture geometries are classified from "closed" to "open" joints according to Bieniawski (1989). The resulting parameter-ranges for fractures in group 2 can be found in table 2. For the numerical fluid flow simulations, we implemented the following workflow: First, we apply a flood-filling algorithm on the initial 3D model along $x$-direction. $R$ and $S$ are calculated on the resulting effective pore space, followed by a numerical permeability estimation, as explained above, in the same direction. Then, we rotate the initial model by $90°$ in the $x-y$ plane, and the procedure explained above is repeated. In this manner, two-directional permeability values for every fracture are obtained, resulting in a total sum of 12800 fluid-flow simulations.

## 3 Results

### 3.1 Hydraulic efficiency

In the following section, we present the results of the numerical fluid flow experiments within the geometries of group 2. For this, we normalize the numerically computed permeabilities ($k_m$) by the permeability predicted by the cubic law ($k_{cl}$) with equivalent mean aperture $\bar{a}$ of the associated effective pore space:

$$\chi = \frac{k_m}{k_{cl}}. \tag{15}$$

Consequently, one can use the hydraulic efficiency $\chi$ as the correction factor in eq. 11 to apply the cubic law to rough fractures. In that way, a fracture whose configuration is close to the parallel plates geometry shows excellent hydraulic efficiency with $\chi$ close to one. In the $R$-$S$-space, the parallel plate fracture configuration exclusively corresponds to a single point with coordinates $(0, 1)$. Perfect hydraulic efficiencies ($\chi = 1$) were validated by flow simulations in parallel-plate fractures. The key result of this study, a model that corrects the cubic law in terms of quantified fracture roughness, is proposed in Fig. 5. Due to the complexity of the results, we fitted a regularized surface with a MATLAB function called "gridfit" (DErrico, 2006) to approximate the solution in $R$-$S$ space. The function can interpolate scattered data within a prescribed bounding box

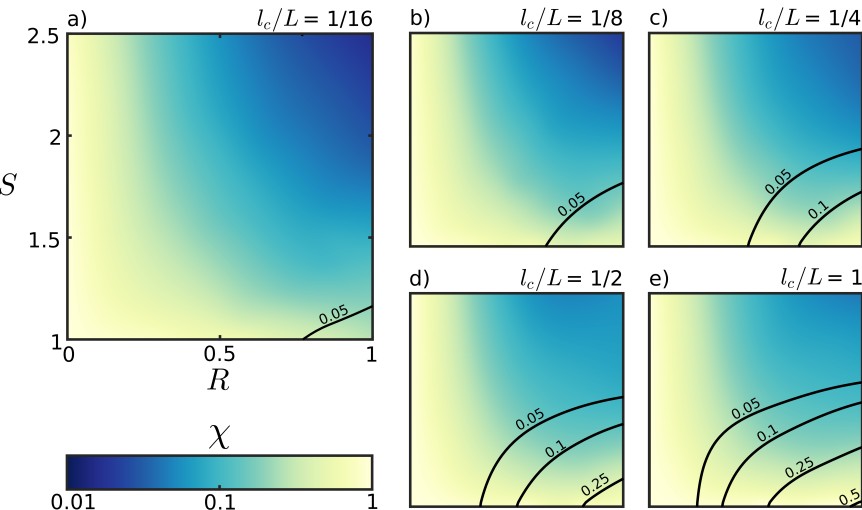

**Figure 5.** The distribution of the hydraulic efficiency $\chi$ for different $l_c/L$ ratios as a function of $R$ and $S$. Both axes limits in *a* correspond with *b-e*. Dark blue colour indicates poor hydraulic efficiency, whereas lighter colour shows increasing accordance with the cubic law. The black contour lines indicate the absolute residuals to the fitted surface (Compare with Fig. 6).

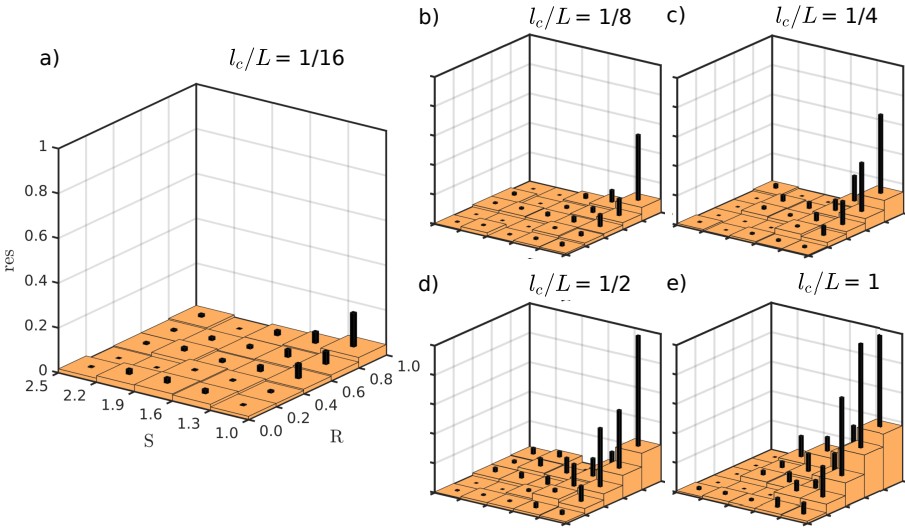

**Figure 6.** Absolute residual values of the fitted surfaces in Fig. 5 for different $l_c/L$ ratios as a function of $R$ and $S$, binned into equally sized boxes. All axes limits in *a* correspond to the ones in *b-e*. Orange boxes indicate the mean absolute residual value of the specific bin, whereas the smaller black boxes on top give the maximum absolute residual.

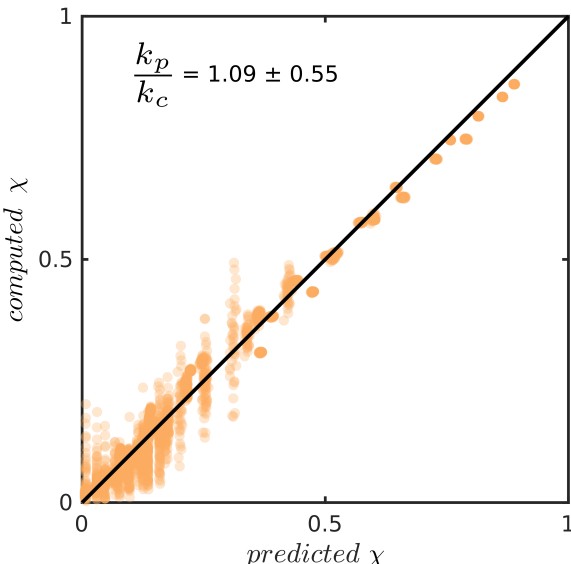

**Figure 7.** Parity plot of predicted versus computed hydraulic efficiency $\chi$ for a total of 2554 fractures with an $l_c/L$ ratio of 1/16. Black line indicates the location of perfect parity. Inlet data gives the mean and standard deviation of predicted ($k_p$) over computed ($k_c$) permeabilities that correspond to all data points in the plot.

and a certain amount of smoothing, resulting in a clearer solution image compared to conventional interpolation techniques. Furthermore, to enable an adequate basis for the fitting, the data was cropped above $R = 1.0$ and $S = 2.5$ to provide sufficient data density in $R$-$S$-space, which reduces the total amount of simulations used for the fitting from 12800 to 10292.

The results display significant deviations from the cubic law approximation, even for fractures where both surfaces are not in contact (i.e., $R \leq 0.23$). We obtain the lowest $\chi$ values in regions of high $R$ and $S$, with a general trend of increasing $\chi$ for larger $l_c/L$ ratios.

To investigate the hydraulic efficiency fluctuations for similar fracture configurations, the absolute residuals of the fitted surface from Fig. 5 to the simulated data are computed. As before, we fit a regularized surface through the scattered points from which we extract the displayed contour lines (Fig. 5). They indicate that the goodness of the fit reduces with increasing $l_c/L$ ratios, especially in the lower right corner. Hence, the non-uniqueness of the data reduces for lower $l_c/L$ ratios, which can be even better seen in Fig. 6. A regional, maximum residual of about 0.2 for fractures with a $l_c/L$ ratio of 1/16 enables a more or less unique parametrization refinement, which is not the case for higher $l_c/L$ ratios. Our finding that low $l_c/L$ ratios show lower reduced variability is consistent with the results of Méheust and Schmittbuhl (2003).

An easy integration of this parametrization refinement into a DFN framework requires a mathematical approximation of the

**Table 3.** Resulting minimal and maximal values for mean aperture ($\bar{a}$), standard deviation of the aperture field ($\sigma_a$), contact fraction ($c$), relative fracture closure ($R$), effective surface area ($S$) and numerical fracture permeability ($k_m$) for the fracture geometries in group 2.

| $l_c/L$ | a | b | c | d |
|---|---|---|---|---|
| 1/16 | 0.0428 | 0.1652 | -0.8226 | 0.8822 |
| 1/8 | 0.6517 | -0.3135 | -0.6751 | 0.6625 |
| 1/4 | 0.9509 | -0.7343 | -0.7852 | 0.7672 |
| 1/2 | 1.0491 | -0.9632 | -1.2065 | 1.1752 |
| 1 | 1.3267 | -1.3174 | -1.6613 | 1.6178 |

fitted surface shown in Fig. 5 a), which was found by the following equation:

$$\chi = 1 - (0.4809 \tanh(0.5139S) + 0.5408) \tanh\left(\frac{R}{39.28 \tanh(-2.451S) + 39.47}\right) \tag{16}$$

To predict single fracture permeability, it is only necessary to know the mean and standard deviation of the aperture field ($\bar{a}$ and $\sigma_a$), the fractional amount of surface contact ($c$) and the surface area protruding into the void space ($sa_f$). From these values, $R$ and $S$ are computed to infer the hydraulic efficiency $\chi$ with eq. 16, which is then multiplied by the permeability predicted by the cubic law with aperture $\bar{a}$ (see eq. 11). Fig. 7 demonstrates the accuracy of eq. 16 to predict hydraulic efficiencies and accompanying permeabilities for fractures with $l_c/L$ ratios of $1/16$. To quantify the hydraulic efficiency fluctuations ($\sigma_\chi$) with respect to its correlation length, we provide a model of the form:

$$\sigma_\chi = (a \times e^R + b)(c \times tanh(S) + d) \tag{17}$$

with corresponding parameter values given by table 3.

### 3.2 Accuracy of the numerical solution

Numerical inaccuracies in solving the Stokes flow equations related to the resolution of the numerical models potentially have an important impact on the results shown here. For numerical permeability estimations of single fractures, the resolution perpendicular to the aperture field is the most crucial part. As the most relevant roughness features are expressed within the uncorrelated region of a fracture (i.e., where $l_c = L$), it is necessary to examine the numerical error introduced due to resolution loss therein. For that, eight fractures with the size of 4096x4096x512 voxels and a $l_c/L$ ratio of $1/16$ are generated in the same manner as explained in section 2.3. For each fracture, 16 subsets are drawn that focus uncorrelated regions of the fracture, resulting in subsets of $256 \times 256 \times 512$ voxels. By this, the fracture part oriented perpendicular to the applied pressure gradient is over-resolved by a factor of two. The resolution of these initial models is then consecutively reduced down to $16 \times 16 \times 32$ voxels (see inlay of Fig. 8 for a workflow sketch) while maintaining a constant dimensional aspect ratio. The resulting permeability at every stage ($k_r$) is then compared to the result at maximal resolution ($k_{max}$), assuming that this represents the most accurate solution. Finally, we compute the error norm according to:

$$||\delta_k|| = \left|\frac{k_r - k_{max}}{k_{max}}\right| \tag{18}$$

Ideally, the error norm should get negligible at the highest resolution. Fig. 8 shows the mean error norm of a total of 128 uncorrelated fracture subsets as a function of voxel size $\Delta r$. A mean error of about $0.01$ % at maximal resolution indicates optimal convergence to the most accurate solution, which validates the numerical procedure. The voxel size of the numerical models used in this study ($0.1\ mm$) results in an acceptable mean error of 7.2 %, as indicated in Fig. 8.

## 4 Discussion

Many studies report that the cubic law increasingly deviates with increasing relative fracture closure (Patir and Cheng, 1978; Brown, 1987; Zimmerman and Bodvarsson, 1996), which is usually attributed to the flow channeling around the contact spots within the fracture, introducing an in-plane tortuosity that reduces the permeability. We quantified the deviations from the cubic law due to vertical roughness features (i.e., amplitudes or Hurst exponents, see Fig. 5). The results suggest that with increasing fracture surface area protruding into the fluid phase, more drag force accumulates at the fluid-matrix interface, which resists the
flow and leads to reduced permeabilities. It is not possible to capture these vertical variations in the flow field with previous 2D modeling approaches (e.g., Patir and Cheng, 1978; Brown, 1987; Renshaw, 1995; Zimmerman and Bodvarsson, 1996), which then results in a biased prediction and the need for more parameters to ensure an adequate quantification. Fig. 9 highlights this issue by computing the norm $||\delta_k||$ between measured and predicted permeabilities for all fractures in this study. With a mean

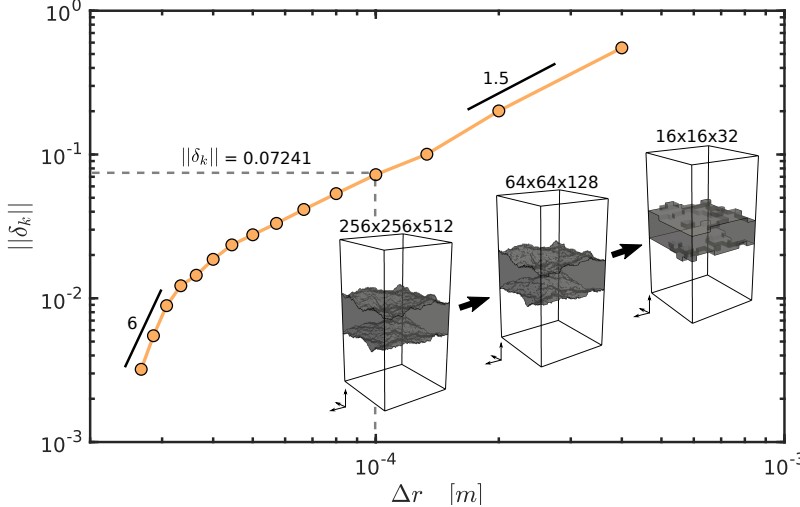

**Figure 8.** Error norm computed by eq. 18 as a function numerical voxel size $\Delta r$. The orange line depicts the mean error of 128 different fracture subsets discretized with decreasing resolutions. The figure inlay sketches the down-sampling procedure is sketched: The maximal resolution is consecutively decreased by 16 in horizontal and 32 voxels in vertical direction (displayed is the maximal, intermediate and minimal reduction stage). Dashed gray lines indicate the voxel size and associated error norm of the numerical simulations used to provide the refined single fracture permeability parameterization. Black lines highlight the convergence rate by showing local slopes as indicated by the attached values.

error of 26.7 % eq. 16 delivers a better prediction compared to the mentioned studies.

As already shown by Méheust and Schmittbuhl (2003), the uncertainty for predicting fracture flow is a function of its $l_c/L$ ratio. Considering flow predictions for uncorrelated fractures (i.e. $l_c/L \geq 1$) is problematic. Blocked pathways connected to the early appearance of the percolation limit (see Fig. 4) or flow enhancing configurations ($\chi > 1$) as also observed by Méheust and Schmittbuhl (2000) are producing substantial variations in their hydraulic efficiencies. With decreasing $l_c/L$ ratios, the impact of vertical flow tortuosity on its permeability increases relative to the impact of in-plane tortuosity, as both start to act

at comparable scales and generally the fractures exhibit larger portions of flow inhibiting regions compared to flow enhancing ones (see Méheust and Schmittbuhl, 2000). On the contrary, the fluctuations in the average flow behaviour decrease significantly with decreasing $l_c/L$ ratios. This suggests that predicting hydraulic properties is constrained to fractures, whose sizes are significantly greater than their correlation lengths. Theoretically, the correlation length is mainly controlled by shear offset and respective gouge generation (Brown, 1995; Méheust and Schmittbuhl, 2000). With the assumption of a perfectly matched

fracture ($l_c = 0$) at its nucleation stage, it is tempting to propose that most natural fractures actually meet the conditions of low $l_c/L$ ratios and subsequently enable the prediction of their hydraulic properties. However, so far, little is known about naturally existing correlation lengths in fractures, as the imaging of in-situ fractures is limited to the size of drill cores. Only Brown (1995) report measurements of $l_c$ by analyzing the power spectral densities of composite topographies for two matched profiles on opposing joint surfaces, shedding some light into their natural ranges. From a mechanical perspective, correlation

lengths that are equal to the size of the fracture seem rather unrealistic, considering that the shear displacement $d_s$ of fractures scales with their length $L_f$ according to $d_s = \alpha L_f{}^{0.5}$ (Schultz et al., 2008). Using $\alpha$ values between 0.01 and 0.001, which is

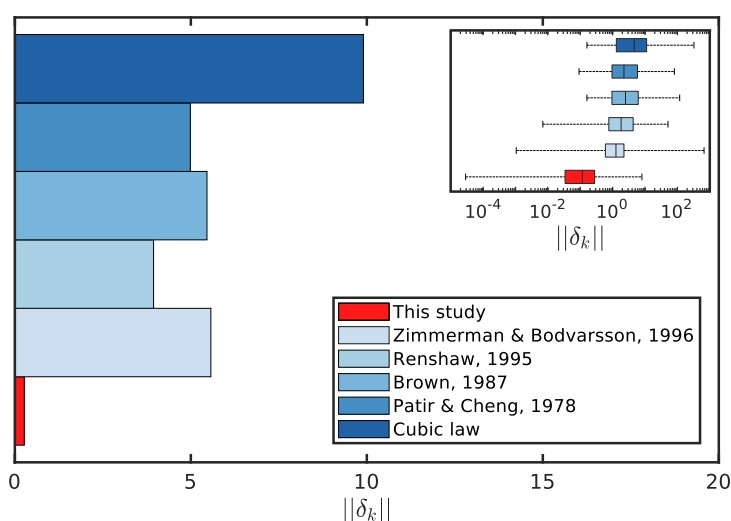

**Figure 9.** Mean error norm $\delta_k$ (see eq. 18) of all fractures considered in this study for different prediction models. The mean error norm recorded for this study is 0.267. Inlet plot shows box and whisker plots incorporating all outliers, i.e. representing minimum and maximum recorded values.

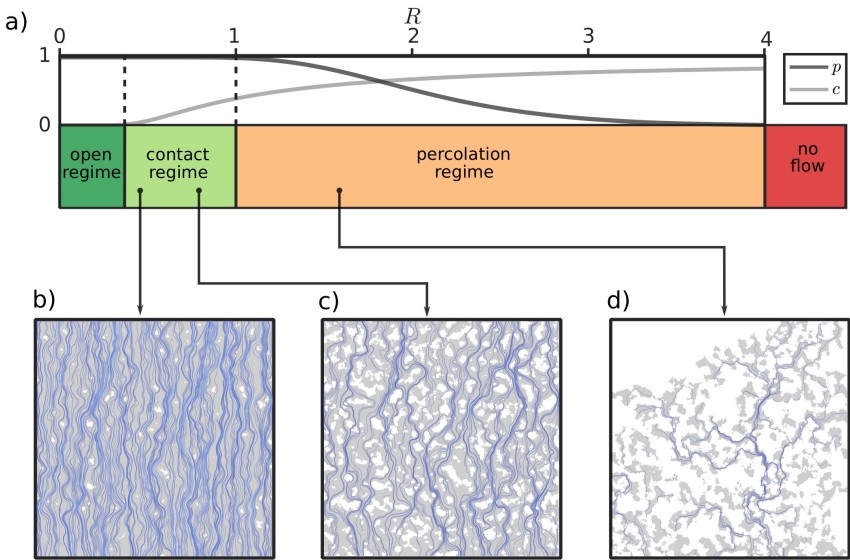

**Figure 10.** A sketch of the three different closure regimes, indicated as a function of $R$ with corresponding $p$ and $c$. The lower part of the figure shows three examples of fluid flow simulations from the indicated regimes. Grey shaded area is fracture void space, whereas white regions indicate contact between both surfaces. Blue lines depict chosen streamlines, approximated from the resulting Stokes velocity vectors.

about the range for Moros' joints in Schultz et al. (2008), results in a maximal $l_c/L_f$ ratio of 0.01. All of that illustrates, that further research on fractures correlation lengths is required because the presence of it is omnipresent in most relevant studies (Brown, 1995; Mourzenko et al., 1996; Méheust and Schmittbuhl, 2003; de Dreuzy et al., 2012; Pyrak-Nolte and Nolte, 2016; Mourzenko et al., 2018).

## 5  Conclusions

To understand the effects of fracture surface roughness on fluid flow, we performed numerical simulations of high-resolution 3D Stokes-flow within fractures for a large synthetic dataset. By consolidating varying asperity amplitudes and roughness scaling within a new quantity, that accounts for the effective increase of surface roughness compared to its parallel plate equivalent, we were able to provide a new way to characterize fracture roughness. By combining the effective surface area with the relative fracture closure, we established a two-parameter characterization scheme that reads similar to a phase diagram. It is utilized to quantify the hydraulic efficiency of single fractures empirically, i.e., the correction factor applied to the current state-of-the-art fracture permeability parametrization (cubic law). Our findings confirm the results of Méheust and Schmittbuhl (2003), and highlight that predicting fracture flow is constrained to scales of at least 16 times larger than the correlation length. The hydraulic efficiency as a function of effective surface area and fracture closure is given by eq. 16, its variability with respect

to the correlation length is given by eq. 17 and table 3, whereas an overall numerical error of 7.2 % has to be considered. Ultimately, we used the percolation probability and contact fractions to classify three different closure regimes that differ in terms of their hydraulic interpretation:

(i)  The open regime defines fractures whose surface walls are not in contact with each other (e.g., unconfined dilatant or karstified fractures). In this regime, we generally observe a good agreement of the cubic law with hydraulic efficiency between $70 - 100\%$ and only extreme roughness configurations (e.g., needle-shaped mineral coatings) result in larger deviations.

    (ii)  The contact regime is characterized by fractures exhibiting a rapidly decreasing hydraulic efficiency from $70 - 10\%$ up to 290     $1$ % in extreme cases, which is caused by strong three-dimensional channeling due to surface roughness and increasing fracture closure. Likely, this regime is most suitable for subsurface conditions, as a certain amount of contact between both fracture surfaces is required to withstand confining pressures.

    (iii)  The percolation regime incorporates fracture configurations that do not percolate at all due to blocked fluid pathways. Here, we do not incorporate fluid flow data, but it is plausible that the hydraulic efficiency is very poor with a maximum 295     of $25\%$, which will quickly converge to $0\%$ due to the effect of decreasing percolation probability with further closure. We observe the no-flow boundary at $R \approx 6$.

Our results generally help to understand the hydraulic response induced by different types of fracture geometries and refine the parametrization of single fracture permeability given by the cubic law. Moreover, the developed quantification scheme allows monitoring and parametrizing the hydraulic and geometric evolution of fractures during aperture field-shaping processes. 300  This parametrization can easily be incorporated in a DFN modeling framework to investigate the hydraulic responses at reservoir scales, assuming that the minimal correlation length is no longer than $1/16$ of the reservoir size. If DFN's of scales close to the correlation length are considered, fluctuations in the average flow behaviour are expected. This can modify network scale flow connectivity and thus requires additional concepts to compute permeabilities (e.g., de Dreuzy et al., 2012).

*Code availability.*  https://bitbucket.org/bkaus/lamem/src/master/ ; commit: 9c06e4077439b5492d49d03c27d3a1a5f9b65d32

*Author contributions.*  MOK wrote the initial draft of the manuscript, performed numerical simulations, analyzed the data and generated the figures. AAP supervised and designed the study, provided the computational framework and edited the manuscript. TSB assisted to data fitting and edited the manuscript. BJPK helped designing the study, assisted to find an analytical model and edited the manuscript.

*Competing interests.*  The authors declare that they have no competing interests

*Acknowledgements.* This work has been funded by the Federal Ministry of Education and Research (BMBF) program GEO:N, grant no. 03G0865A. The authors gratefully acknowledge the computing time granted on the supercomputer Mogon II at Johannes Gutenberg University Mainz (hpc.uni-mainz.de). The authors sincerely thank Guido Blöcher and an anonymous referee for reviewing this manuscript.

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
