# Peer review of "The hydraulic efficiency of single fractures: Correcting the cubic law parameterization for self-affine surface roughness and fracture closure"

_Solid Earth, 2019_

## Referee Comment (RC1) · Guido Blöcher (Referee) · 14 Feb 2020

Review of the manuscript "The hydraulic efficiency of single fractures: Correcting the cubic law parameterization for self-affine surface roughness and fracture closure" / https://doi.org/10.5194/se-2019-190

In the presented study, a non-dimensional fracture roughness quantification-scheme is acquired, opposing effective surface S area against relative fracture closure R. This is used to capture deviations from the cubic law as a function of quantified fracture

roughness, here termed hydraulic efficiencies.

The paper presents the approximation of a correction factor for the cubic law depending on S and R. Since S and R are only depending on geometrical properties of the fracture surface (which could be determined by real measurement), the presented approach for correcting the cubic law parameterization seems highly applicable.

Besides the scientific significance the paper is well written, structured, and the reference list is adequate. I suggest accepting the paper after minor revisions.

General comments: The general focus of the paper is to present a correction factor for the cubic law in the R-S space. This by its own is of particular interest. In contrast, all simulations were performed with a pressure difference of 0.01 Pa and laminar flow conditions are assumed. This opens the question, are the correction factors the same (in R-S space) for other pressure gradients and therefore other flow velocities.

Although, a non-dimensional fracture roughness quantification-scheme is acquired, the authors could indicate the dimensions of their simulations. This would help to understand at which scale the correction scheme is applicable. Values of length, height, mean aperture, etc. should be mentioned either as absolute or relative values.

This comment is the most important one. As mentioned before the presented approach seems highly applicable. Unfortunately, the proof with real data is missing. It would be nice to use, e.g. laboratory measurements, to proof the presented approach. If no data are available, we could provide surface scans with a resolution of 50 $\mu$m and related fracture permeability measurements. Furthermore, aperture distribution, mean aperture, contact area are known. The authors could show how they determine the S-R-values and can subsequently compare the corrected permeability values with the measurements. (contact: Guido Blöcher, bloech@gfz-potsdam.de)

Specific comments: P1L2: Replace "Yet" with "In contrast".

P1L9: Here a sentence regarding the dimension of the simulation (length, width, and

aperture) should be added.

P3L75: What means: low Reynolds numbers and low-pressure gradients? The authors should provide ranges where the presented approach is applicable.

P3L81: What is effective fluid viscosity? $\mu$ is the dynamic fluid velocity.

Figure 2: The vertical axis has no axis label. It seems it indicates the aperture. Since the aperture is provided by the color code, a 2D image would be sufficient.

P6L126: Often $\varphi$ indicates the porosity but not here. I suggest using another symbol than $\varphi$ to indicate the correction factor.

P6L128: For S = 1 and R = 0 we would mimic flow between parallel plates. For this situation, the correction factor should be one and the cubic law could validate the simulation. This validation was done but is somehow hidden in figure 5. The authors should emphasis that this simple check was performed.

P6L135: Why a representation of the matrix is required? Only the fracture is simulated and the boundary condition is a non-slipping boundary. It is not clear why the complete matrix-fracture-system is considered.

P6L136-138: The sentence seems to be incomplete and should be rewritten. Furthermore, the macroscopic flow direction (y-direction) should be mentioned.

P6L141: This is an integration of v_y over the total volume and not a volume integral. Since v_x and v_z should be small compared to v_y these quantities could be used to determine v ÌĚ.

P6L144: The dynamic viscosity is denoted with $\eta$ before it was $\mu$.

Table 1: The number of parameter combinations for group 1 is given to be 400. If I multiply the n_g1 (4*4*4*5) it should be 360. Furthermore, the fracture configurations for group 1 should be 360*r_g1=7200 but only 6400 are mentioned in the table caption.

P8L1: Again, values about the fracture aperture and the representation by the voxels is missing.

P8L173: 12800 flow simulations are mentioned assuming 6400 fracture configurations. If comment 12 is corrected or explained the authors should revise the number of flow simulations (14400).

P9L194: In case the authors decided for another symbol than $\varphi$, they should correct the symbol here.

Figure 5: The figure is fine in color mode but almost no difference between blue and red is visible in greyscale mode. Maybe, the color scheme can be adjusted.

P13L240: Please add a space after closure.

Figure9: The mean error norm was obtained for the considered fractures and applied boundary conditions. The authors should discuss if this error norm changes for other flow regimes regarding Reynold number, flow velocity, pressure gradients, etc..

---

## Referee Comment (RC2) · Anonymous Referee #2 · 23 Mar 2020

This manuscript addresses the hydraulic behavior of geological fractures using a numerical approach. The authors perform low Reynolds CFD inside synthetic fractures and analyse the impact of the fracture closure R and Hurst exponent of the fracture walls, H, on the fractures' permeability. They introduce an effective surface area S which accounts for the change in surface area when changing the Hurst exponent H, and therefore replaces H in the parametrization. They then compute the mean behavior of a large population of geometrically equivalent fractures (i.e., fractures generated with the same geometrical parameters) as a function of R and S. They also discuss how the

value of the correlation length impacts the variability of the hydraulic efficiencies within the population, and show that for the smallest investigated correlation length that variability is negligible. In this case they fit an analytical formula to the dependence of the hydraulic efficiency on R and S and propose that this formula be used as an analytical model in reservoir scale DFN models.

Few studies have so far examined the role of the correlation length, though a previous work (using a Reynolds equation based approach) has shown that it strongly impacts the hydraulic behavior of fractures, but also the variability of hydraulic efficiencies of fractures of identical statistical geometrical parameters. In addition, few studies have addressed systematically the impact of the Hurst exponent. The paper is well written and easy to read. The methods are sound and the interpretation overall convincing. I therefore recommend publication.

I provide below a number of comments which may have to be addressed prior to publication.

1) Main comments:

* Large closures and percolation analysis:

I don't think it makes much sense to investigate closures R much larger than 1. Indeed, the hypothesis of perfect plastic closure (overlapping regions just disappear) is not too bad for configurations in which a moderate proportion of the fracture plane is closed (i.e., for R $<\sim$ 1). But for larger closures one would expect the real geometry to be significantly different from that obtained with this crude approximation.

I turns out that the results relative to the hydraulic efficiency are shown only for R < 1. The study of percolation, on the contrary, is only interesting for R > 1 since the percolation probability starts taking values strictly smaller than 1 precisely for these configurations of large closure (R > 1). This section is therefore, in my opinion, rather irrelevant. I would suggest to remove it.

* Figures 5 and 6, and the corresponding discussion (pages 10 and 11):

Instead of showing a plot that is interpolated from the raw data using a Matlab function whose principle is not explained and whose parameters are not given, I would suggest that the authors perform their own box-averaging to show local mean values of S as a function of S and R, but also that they also provide similar information for the fluctuations of the statistics, for example in terms of the standard deviations of values within various (R,S) ranges. Such a figure coud be added following the model of Figure 5, and would complement it.

In a way the information provided in Figure 6 contains this type of information, but in a less straightforward manner, and though the interpretation provided by the authors is correct, the choice of words matters. This is not about the "accuracy of the presented model", this results from the fact that the model corresponds to the average behavior of a population, and that fluctuations in hydraulic are found within the population. These fluctuations are all the larger as the correlation length is larger. The model may be very accurate for the average behavior (and probably is). And the authors could provide a model for the standard deviation around the mean behavior by fitting the data of the figure I am suggesting above.

Similarly I think that the wording used in the sentence of page 15, line 276, is misleading when mentioning "a prediction error of 26.7%".

* I am not quite sure I fully understand how the test on the accuracy of the numerical solution is done.

Firstly, the notion of "uncorrelated part of a fracture" is strange to me, as the uncorrelated vs. correlated feature is a question of scale rather than location. Perhaps it is simply a question of formulation. Similarly, the sentence of line 228, "16 subsets are drawn that focus on the uncorrelated parts of the fractures that corresponds to ..." is not clear to me.

* Last sentence of the paper: "This parametrization could easily be incorporated in a DFN modeling framework to investigate the hydraulic response at reservoir scales".

Yes, it could. It could be interesting to substantiate, though, for two reasons.

First, this model is obtained for fractures of correlation length L/16. Does it still hold whatever the correlation length if it is smaller than L/16 ? If not the model could only be used in models of fractured reservoirs for which all fractures exhibit a ratio L/lc = 16.

Second, the hydraulic behavior of DFN of rough fractures is not necessarily properly described by that of a DFN of parallel plate fractures of suitably adjusted apertures. There can be coupling between fracture scale heterogeneity and network-scale heterogeneity, that is, fracture scale flow heterogeneity can in some cases modify the flow connectivity at the network scale. However de Dreuzy et al (JGR 2012) have shown that this can only occur if the correlation length is not significantly smaller than one or two tenths of the medium size. At reservoir scale this is clearly never the case. But this is not trivial and could be discussed.

2) Various comments on other points along the text:

* The introduction is rather short, but logically organized, and provides a proper summary of the state of the research on the topic so far. The authors use an approach relying on a large statistics of fracture with identical. They could mention that the first approach of this kind was proposed by Méheust and Schmittbuhl in a JGR paper in 2001, studying populations of synthetic rough fractures with self-affine aperture fields (that is, for lc/L=1).

* In the presentation of Eq. (1), the hypothesis of permanent flow is missing.

* In Eq. (6), the mathematical notation is strange: a is used both for the aperture field prior to negative values being put to 0, and for the the aperture field whose negative values have been put to 0. Of course when coding one may use the same variable name, overwriting the previous variable a, but mathematically they are two different

quantities.

* Page 5, line 104: "leaving $H$ as a measure for the intensity of small scale roughness".

This explanation is a bit caricatural. $H$ is rather a measure of the ratio between larger scale roughness and smaller scale roughness (the ratio being always larger than 1 since h>0, but all the smaller as H is smaller).

* Equation (8): some authors choose to divide the standard deviation of the aperture by the mechanical aperture, which is the mean aperture prior to putting negative apertures to 0 and thus corresponds to the distance between the mean planes of the facing topographies. Is there a particular reason why you chose to use the mean aperture ?

* Page 4, line 113: why don't you express the condition of contact in terms of R (R >= 1/(3\sqrt{2}) ?

* Page 5, line 121: It seems that a simple way of presenting S would be as the ratio of the fracture surface's area to twice that of its projection on the fracture plane.

* Page 6: was the conservation of the total volumetric flow rate tested ? What are the relative flow rate fluctuations between all sections transverse to the mean flow ?

* Table 1: It would be interesting to have the mininum and maximum values of R in the table.

* Page 8, line 184-185, about the inset plot: the contact fraction is only controlled by the PDF of apertures prior to setting negative values to 0; that PDF is mostly independent of lc/L (though if one looks closely one may find a slight dependence), and therefore only dependent on the fracture closure. This is well known.

* Page 12, line 225: Here you probably mean "perpendicular to the fracture plane", i.e., the vertical direction if the fracture is horizontal.

* Equation (8): in this equation, it seems that the norm is simply the absolute value of

the relative error. Why use a square inside a root mean square ?

* Discussion of page 14: here you could mention that the lower the value of lc/L, the larger the impact of the vertical flow tortuosity on the fracture's permeability.

* Page 15, line 262: "correlation lengths that are equal to the size of the fracture .... seem rather unrealistic".

The origin of the correlation length is not generally known, is it ? Is it mechanical ? A fresh fracture without shift along the fracture plane would present a constant aperture field, one with a shift of length l would have a correlation length lc = l in that direction, but then the aperture field would be anisotropic.

* Page 15, line 269: Here and elsewhere I would use "parallel plate equivalent" (which refers to the geometry) rather than "cubic law equivalent", which involves a hydraulic concept. The two fractures are equivalent in that their mean apertures are identical (a geometric feature), not in that their hydraulic behavior is the same (this equality defines the fracture's hydraulic aperture).

3) Writing:

The paper is overall very well and clearly written. Here are a few corrections that could be made:

* Shouldn't the vectorial quantities (including \nabla) appear in bold fonts ?

* Page 4, line 100: I would call the "rescaling factor" simply a "prefactor".

* Page 6, line 137: ")" should be removed after "0.01 Pa".

* Page 6, line 145: here I'd write "with $\eta$ the fluid's dynamics viscosity".

* Page 8, line 176: I think "build" should be "built" here; please check.

* Page 12, line 219: "multiplied by" rather than "on".

* Page 12, line 225: "the resolution perpendicular to the flow direction".

---

## Author Comment (AC1) · 8 Apr 2020

We sincerely thank Guido Blöcher for his review. His comments were very useful and helped us to improve the quality of our manuscript. Please find below a point-by-point response to the general and specific comments (comments of the reviewer in black and our response in red, text changes appear in italic font) .

On behalf of all authors, Yours sincerely,

[Figure]

Maximilian Oskar Kottwitz

**General comments:**

The general focus of the paper is to present a correction factor for the cubic law in the R-S space. This by its own is of particular interest. In contrast, all simulations were performed with a pressure difference of 0.01 Pa and laminar flow conditions are assumed. This opens the question, are the correction factors the same (in R-S space) for other pressure gradients and therefore other flow velocities.

Laminar flow conditions are essential for using the Stokes equations instead of the Navier-Stokes equations and are valid for Reynolds numbers below unity (see Zimmermann & Bodvarsson, 1996). Above that, inertial forces start to introduce turbulences which further reduce overall flowrates. By choosing a constant pressure difference of 0.01 Pa, we ensured that the maximal resulting Reynolds numbers is below unity. In figure 1, we show the Reynolds number Re (computed by $Re = \frac{\rho \bar{v} \bar{a}}{\mu}$, with fluid density $\rho$, volume average velocity $\bar{v}$, mean aperture $\bar{a}$ and fluid viscosity $\mu$) as function of mean aperture $\bar{a}$. As evident from the picture, the bulk of the simulations range from 10-3 to 10-8. Which means, for sub-millimeter apertures, the pressure gradient could be larger than 0.01 Pa without interfering with the unity threshold. For sub-centimeter apertures, we reach the unity threshold, so strictly speaking, this poses an upper limit for the applicability of the computed correction factors. For pressure differences lower than 0.01 Pa, we generally assume the correction factors to be valid. If fractures with larger apertures (e.g. cm range) are considered, the applied pressure gradients need to be reduced such that they fulfill the above-mentioned criteria of Reynolds numbers

below unity. Yet, if laminar flow conditions exist, the proposed correction factors remain applicable.

Although, a non-dimensional fracture roughness quantification-scheme is acquired, the authors could indicate the dimensions of their simulations. This would help to understand at which scale the correction scheme is applicable. Values of length, height, mean aperture, etc. should be mentioned either as absolute or relative values.

We indicated a fixed physical voxel size of 0.1 mm. With a constant model resolution of 512*512*128 voxels, this results in a model size of 51.2*51.2*12.8 mm. Table 1 gives input metrics used to generate the synthetic fractures. The resulting range of mean apertures ($\bar{a}_{min} = 1.910 * 10^{-04}\ m; \bar{a}_{max} = 4.961 * 10^{-03}\ m$) as well as absolute permeability values ($k_{min} = 6.786 * 10^{-14}\ m^2; k_{max} = 7.940 * 10^{-07}\ m^2$) were inserted a new table (table 2) at page 9.

This comment is the most important one. As mentioned before the presented approach seems highly applicable. Unfortunately, the proof with real data is missing. It would be nice to use, e.g. laboratory measurements, to proof the presented approach. If no data are available, we could provide surface scans with a resolution of 50 $\mu m$ and related fracture permeability measurements. Furthermore, aperture distribution, mean aperture, contact area are known. The authors could show how they determine the S-R-values and can subsequently compare the corrected permeability values with the measurements. (contact: Guido Blöcher, bloech@gfz-potsdam.de)

The validation of the proposed concept with real data is truly of high interest. In fact, we are already preparing follow-up studies targeting exactly this:
In one study, we validate the results of the numerical simulations by using 3D printed

versions of some fracture realizations generated in this study and perform laboratory experiments to obtain their permeabilities to subsequently compare them to their numerical equivalents.

In another study, we test the performance of the proposed correction scheme to predict permeabilities of real fractures by utilizing CT-scans of those (partly obtained from www.digitalrocksportal.org and partly self-acquired – by now a total of 32 CT-scans). It also incorporates simple pre-processing advices to prepare the CT data for numerical simulations and the computation of the S-R-values, which we by default calculate from voxel datasets.

We genuinely appreciate the offer of providing data, especially the fracture permeability measurements are of high interest. However, to fully capture the statistics of a fracture's aperture field (specifically in terms of its correlation length and contact areas) we require volumetric (voxel-based) data, rather than separated scans of fracture surfaces. By that, we hope to reflect the true matching of the opposing fracture surfaces, which determines the correlation length of the aperture field. Furthermore, we computed the S-R-values from volumetric data.

We fully understand the request for a proof with real data, however we think that this is beyond the scope of this methodological study and we think that it is more appropriate to deliver this in future studies.

**Specific comments:**

P1L2: Replace "Yet" with "In contrast".

Incorporated suggest change at indicated location.

P1L9: Here a sentence regarding the dimension of the simulation (length, width, and aperture) should be added.

We changed the sentence at P1L10 to:
*Each fracture consists of two random, $25cm^2$ wide self-affine surfaces with prescribed roughness amplitude, scaling exponent, and correlation length, which are separated by varying distances (mean apertures in submillimeter range) to form fracture configurations that are broadly spread in the newly formed two-parameter space.*

P3L75: What means: low Reynolds numbers and low-pressure gradients? The authors should provide ranges where the presented approach is applicable.

We changed the sentence at P3L75 to:
*Assuming, that the flow is solely laminar (Reynolds numbers below unity according to Zimmermann and Bodvarsson, 1996), the fluid viscosity is constant and . . .*

P3L81: What is effective fluid viscosity? $\mu$ is the dynamic fluid viscosity.

We simplified the sentence at p3L81 to:
*. . . with the fluid's dynamics viscosity $\mu$, . . .*

Figure 2: The vertical axis has no axis label. It seems it indicates the aperture. Since the aperture is provided by the color code, a 2D image would be sufficient.

Technically, a 2D image would be sufficient as vertical axis and colorbar provide the same information. However, we choose the 3D representation to put emphasis
on the effective surface area increase caused by a lower Hurst exponent for equal mean and standard deviations. In 2D, we had the impression that this was not as straightforward to see. On top of that, the necessary visual comparison of mean and standard deviation by the black solid and dashed lines could not be realized in 2D.

This is why we would rather keep the Figure as is, while changing the caption as follows:

*Two aperture fields constructed from synthetic fractures. Both aperture fields are based on the same sets of random numbers with varying Hurst exponents H, which is a) 0.4 and b) 0.8. The two statistical parameters $\bar{a}$ and $\sigma_a$ are indicated by black solid and dashed lines, respectively. Axis units are in mm, while the vertical axis (indicating aperture) is exaggerated by a factor of two for clarity. Note that $\bar{a}$ and $\sigma_a$ are identical for a) and b). Increasing height fluctuations at smaller scales, caused by a lower Hurst exponent results in a larger effective surface area S for fracture a) compared to b).*

P6L126: Often $\phi$ indicates the porosity but not here. I suggest using another symbol than $\phi$ to indicate the correction factor.

We decided to change the symbol from $\phi$ to $\chi$ throughout the paper, as we agree that the use of $\phi$ can lead to confusion.

P6L128: For S = 1 and R = 0 we would mimic flow between parallel plates. For this situation, the correction factor should be one and the cubic law could validate the simulation. This validation was done but is somehow hidden in figure 5. The authors should emphasis that this simple check was performed.

We added the following sentence at P9L197:

*Perfect hydraulic efficiencies ($\chi = 1$) were validated by flow simulations in parallel-plate*

*fractures.*

P6L135: Why a representation of the matrix is required? Only the fracture is simulated and the boundary condition is a non-slipping boundary. It is not clear why the complete matrix-fracture-system is considered.

Numerically, we don't discretize the matrix. We zero out the corresponding entries in the jacobian matrix to gain computation time. However, we ensure constant spatial extents for all simulations for the volume integration of the velocity, such that all permeabilities are normalized to the same fracture volume.

P6L136-138: The sentence seems to be incomplete and should be rewritten. Furthermore, the macroscopic flow direction (y-direction) should be mentioned.

We changed the sentence at P6L136-138 to:
*Different pressures are applied on two opposing boundaries ($\Delta P = 0.01$ Pa for all models), while the remaining boundaries are set to no-slip. This fixes the principal direction of fluid movement (here it is in y-direction, e.g. Fig. 3).*

P6L141: This is an integration of $v_y$ over the total volume and not a volume integral. Since $v_x$ and $v_z$ should be small compared to $v_y$ these quantities could be used to determine v ÌE.

For directional permeabilities, we are just interested in the $v_y$ component for the applied boundary conditions (no-slip at boundaries next to pressure boundaries). Considering of $v_x$ and $v_z$ would only be important, if we open the no-slip boundaries

(e.g. apply a linear gradient between opposing fracture boundaries).
We updated the sentence at P6L141:
*After ensuring that the numerically converged solution is obtained (see appendix A in Eichheimer et al., 2019), the velocity component parallel to the principal flow direction is integrated over the volume to compute the volume average velocity $\bar{v}$ according to:*
. . .

P6L144: The dynamic viscosity is denoted with $\eta$ before it was $\mu$.

We changed the viscosity notation to $\mu$.

Table 1: The number of parameter combinations for group 1 is given to be 400. If I multiply the $n_{g1}$ (4*4*4*5) it should be 360. Furthermore, the fracture configurations for group 1 should be 360*$r_{g1}$=7200 but only 6400 are mentioned in the table caption.

Apologies, we indeed forgot to update the total number of parameter combinations to 320 (4*4*4*5). This results in 6400 geometries and 12800 flow simulations.

P8L157: Again, values about the fracture aperture and the representation by the voxels is missing.

We expanded the sentence at P8L157 by:
*. . . with a fixed physical voxel size of 0.1 mm, resulting in a model domain of 51.2 x 51.2 x 12.8 mm.*

We added the following at P8L166:
*. . ., yet they all remain fully percolating (resulting mean apertures range from 0.15 to 4.96 mm).*

P8L173: 12800 flow simulations are mentioned assuming 6400 fracture configurations. If comment 12 is corrected or explained the authors should revise the number of flow simulations (14400).

See comment above.

P9L194: In case the authors decided for another symbol than $\phi$, they should correct the symbol here.

See comment above. Symbol was also changed in Figure 5 and Figure 7.

Figure 5: The figure is fine in color mode but almost no difference between blue and red is visible in greyscale mode. Maybe, the color scheme can be adjusted.

We revised the figure with a grey-scale friendly and also sequential colorbar (see figure 2), which makes more sense than a diverging one. Colour descriptions in the figure caption are changed accordingly.

P13L240: Please add a space after closure.

We incorporated suggest change at indicated location.

Figure9: The mean error norm was obtained for the considered fractures and applied boundary conditions. The authors should discuss if this error norm changes for other flow regimes regarding Reynold number, flow velocity, pressure gradients, etc..

Increasing the pressure gradient results in higher flow velocities, ultimately leading to larger Reynolds numbers. As discussed above, the results are only valid, if the Reynolds number is below or equal to unity, i.e. laminar flow conditions are present, which is a fundamental assumption of using the Stokes equations. Quantifying errors in flow regimes other than laminar would require solving the full Navier-Stokes equations, which is beyond the scope of this study but poses an interesting challenge for future work.
* * *
[Figure]

[Figure]

**Fig. 1.** Computed Reynolds number as function of mean aperture for different correlation-length-to-size ratios

[Figure]

**Fig. 2.** The distribution of the hydraulic efficiency for different correlation-length-to-size ratios as function of R and S

---

## Author Comment (AC2) · 8 Apr 2020

We sincerely thank the referee for reviewing this manuscript. His/her constructive and well-structured comments helped us to advance our manuscript. Please find below a point-by-point response to the referee comments (comments of the reviewer in black and our response in red, text changes appear in italic font)

On behalf of all authors, yours sincerely,

[Figure]

Maximilian Oskar Kottwitz

**Main comments:**

* Large closures and percolation analysis:
I don't think it makes much sense to investigate closures R much larger than 1.
Indeed, the hypothesis of perfect plastic closure (overlapping regions just disappear)
is not too bad for configurations in which a moderate proportion of the fracture plane
is closed (i.e., for $R \leq 1$). But for larger closures one would expect the real geometry
to be significantly different from that obtained with this crude approximation. It turns
out that the results relative to the hydraulic efficiency are shown only for $R < 1$. The
study of percolation, on the contrary, is only interesting for $R > 1$ since the percolation
probability starts taking values strictly smaller than 1 precisely for these configurations
of large closure ($R > 1$). This section is therefore, in my opinion, rather irrelevant. I
would suggest to remove it.

The main purpose of the percolation analysis was to narrow down the parameter
space for the presented generation-procedure of synthetic fracture models for the
fluid-flow simulations. On top of that, we also wanted to know if there is a dependency
of percolation and contact fraction on the effective surface area of the fractures. Since
there was no notable dependence on S, one could have already excluded this from
the paper, but since we use parts of the data for our conceptual model in figure 10, we
initially decided to keep it.
We are fully aware that realistic geometries with configurations of $R > 1$ should differ
significantly from the synthetic geometries we generated here, and we agree that
adding the percolation analysis to the results section could be misleading due to the

points you mentioned. We therefore now shift the figure to the description of the synthetic dataset (section 2.3), as it potentially shows closure behavior in a statistical manner. We clarified in the text that this approximation is rather crude for fractures with $R > 1$ but necessary to get an idea of the boundaries of the parameter space, which ultimately helped us to synthesize our conceptual model in the discussion.

This led to a few changes in the paper:
Section 3.1 was removed and parts of it were inserted into section 2.3. Furthermore, we exchanged the names of group 1 and 2 (group 2 are now the fluid flow geometries) to be chronologically consistent.

In addition, we added this line to address your comment:
*Following this, we have chosen to limit the fracture geometries for the fluid flow simulations to configurations with $R \leq 1$ to (i) exclude non-percolation systems and (ii) limit the effect of the above-mentioned "melting" hypothesis, which intensifies with increasing R.*

We also include a new table (table 2) on page 9, indicating the value ranges of mean aperture, standard deviation of aperture, contact fraction, R and S as well as effective permeability for all models in group 2, as this was also requested in by Reviewer 1.

\* Figures 5 and 6, and the corresponding discussion (pages 10 and 11):
Instead of showing a plot that is interpolated from the raw data using a Matlab function whose principle is not explained and whose parameters are not given, I would suggest that the authors perform their own box-averaging to show local mean values of S as a function of S and R, but also that they also provide similar information for the fluctuations of the statistics, for example in terms of the standard deviations of values within various (R,S) ranges. Such a figure coud be added following the model of

Figure 5, and would complement it.

In a way the information provided in Figure 6 contains this type of information, but in a less straightforward manner, and though the interpretation provided by the authors is correct, the choice of words matters. This is not about the "accuracy of the presented model", this results from the fact that the model corresponds to the average behavior of a population, and that fluctuations in hydraulic are found within the population. These fluctuations are all the larger as the correlation length is larger. The model may be very accurate for the average behavior (and probably is). And the authors could provide a model for the standard deviation around the mean behavior by fitting the data of the figure I am suggesting above.

Presenting experimental data with that amount of complexity is always non-trivial. We have chosen the open-source "gridfit" routine (using a smoothening factor of 4 and their default interpolation settings, i.e. Delaunay triangulation) because we already have been using it for a while due to its convenience and found that it was also used in a few other publications. However, we agree that this might seem unintuitive in comparison to using box-averaging and standard fitting routines, although it delivers comparable results. To be on the safe side, we performed the box-averaging on the raw data as suggested above to obtain local mean and standard deviations (see figure 1 and 2). Here, we have chosen the box-sizes such that they always contain 20 or more data points. If this condition does not hold, the boxes were left blank on the plot. We could integrate the new plots with the box-averaged data into the paper or use the visually more appealing interpolated "gridfit" data as already present. There, the fluctuations of the hydraulic efficiency are indicated by the black contour lines. We prefer using the original figure, as we find that it generally shows the trends of the data quite well and is easier to interpret as it combines the average behavior and the fluctuations.

We were not aware that a model for the standard deviation would be helpful using this kind of model – thank you for pointing it out. To deliver that, we used the box-averaged

standard deviation data, took the center points of each non-empty box and assigned the corresponding value to perform standard surface fitting. We found that functions of the form $(a * exp(R) + b) * (c * tanh(S) + d)$ gave reasonable approximations of the hydraulic efficiency fluctuations, whereas the fitting parameters change for different correlation lengths. For that we generated a new table (table 3, page12), containing the fitting coefficients.

We changed the sentence on page 11, line 207 to:
*To investigate the hydraulic efficiency fluctuations for similar fracture configurations,*
*...*

Add the end of the section, page 12 line 221, we added the following (note that we changed $\phi$ to $\chi$ throughout the text to prevent confusion with porosity - see RC1):
*To quantify the hydraulic efficiency fluctuations ($\sigma_\chi$) with respect to its correlation length, we provide a model of the form:*
$\sigma_\chi = (a \times e^R + b)(c \times tanh(S) + d)$
*with corresponding parameter values given by table 3.*

Similarly I think that the wording used in the sentence of page 15, line 276, is misleading when mentioning "a prediction error of $26.7\%$.

We also changed the wording at 15, line 276 to:
*The hydraulic efficiency as a function of effective surface area and fracture closure is given by eq. 16, its variability with respect to the correlation length is given by eq. 17 and table 3 whereas an overall numerical error of $7.2\%$ has to be considered.*

Accordingly, we had to change the abstract at page 1, line 20 to:
*An equation was provided that predicts the average behavior of hydraulic efficiencies*

*and respective fracture permeabilities as a function of their statistical properties. A model to capture fluctuations around that average behavior with respect to their correlation lengths has been proposed. Numerical inaccuracies were quantified with a resolution test, revealing an error of $7.2\%$.*

\*I am not quite sure I fully understand how the test on the accuracy of the numerical solution is done.

The numerical estimation of permeabilities is resolution dependent, i.e. it requires sufficient level of discretization to obtain the correct result. To investigate the impact of this resolution dependency and subsequently the accuracy of the numerical solution at a certain level of discretization, proper resolution tests have to be conducted. In praxis, the same model is discretized with increasing resolutions and usually the resulting permeability converges to a constant value with respect to increasing resolutions – this is then thought to be the true solution. From this kind of calibration curves it is possible to estimate the numerical error at a certain level of discretization.
In our case, we first run simulations in several fractures with lc/L ratios of 1 at large resolutions and then subsequently reduce the resolution of the same models and investigate how the permeability fluctuates (quantified with the error norm in figure 8 as a function of voxel size, i.e 1/resolution).

Firstly, the notion of "uncorrelated part of a fracture" is strange to me, as the uncorrelated vs. correlated feature is a question of scale rather than location. Perhaps it is simply a question of formulation. Similarly, the sentence of line 228, "16 subsets are drawn that focus on the uncorrelated parts of the fractures that corresponds to ..." is not clear to me.

With "uncorrelated part of a fracture" we mean a smaller portion of the fracture where both fracture surfaces are fully uncorrelated. This can be any region of the fracture, with size equal to the correlation length, regardless of its location. The main surface features that inhibit flow are expressed in those regions, as above the correlation length the fracture is more or less planar. But since we simulated fractures of equal spatial extents with varying correlation lengths, we needed to quantify the numerical error caused by the loss of resolution in exactly those locations.

E.g: Our fractures are resolved with 512x512x128 voxels. For a fracture with lc/L of 1, the uncorrelated surface features are highly resolved with 512x512. In contrast to that, the uncorrelated regions of a fracture with lc/L of 1/16 are only resolved with a resolution of 32x32. The grey-dashed lines in figure 8 highlight the numerical error ($7.2\%$) that is connected to the resolution we have chosen for our fractures with lc/L ratios of 1/16. For increasing lc/L ratios this error should reduce, as the uncorrelated regions become higher resolved.

In order to clarify this, we changed the wording on page 12, line 226 to:
*As the most relevant roughness features are expressed within the uncorrelated region of a fracture (i.e., where $l_c = L$), . . .*

We also changed line 226-228 to:
*For that, eight fractures with the size of $4096x4096x512$ voxels and a $l_c/L$ ratio of $1/16$ are generated in the same manner as explained in section 2.3. For each fracture, 16 subsets are drawn that focus uncorrelated regions of the fracture, resulting in subsets of $256x256x512$ voxels.*

\* Last sentence of the paper: "This parametrization could easily be incorporated in a DFN modeling framework to investigate the hydraulic response at reservoir scales".

Yes, it could. It could be interesting to substantiate, though, for two reasons.

First, this model is obtained for fractures of correlation length L/16. Does it still hold whatever the correlation length if it is smaller than L/16 ? If not the model could only be used in models of fractured reservoirs for which all fractures exhibit a ratio L/lc = 16.

Yes, you are right. Equation 17 can only be used for fractures in systems that are at least 16 times larger than the minimal correlation length. However, we can now use the model for the standard deviation of hydraulic efficiencies to incorporate fluctuations for systems that are closer to the correlation length. See comment below for changes in the paper.

Second, the hydraulic behavior of DFN of rough fractures is not necessarily properly described by that of a DFN of parallel plate fractures of suitably adjusted apertures. There can be coupling between fracture scale heterogeneity and network-scale heterogeneity, that is, fracture scale flow heterogeneity can in some cases modify the flow connectivity at the network scale. However de Dreuzy et al (JGR 2012) have shown that this can only occur if the correlation length is not significantly smaller than one or two tenths of the medium size. At reservoir scale this is clearly never the case. But this is not trivial and could be discussed.

Thank you for pointing that out. If DFN with spatial extents close to the fracture's correlation length are considered, fluctuations in the flow properties have to be taken into account. The paper you mentioned demonstrates nicely how this can be done.

We added the following sentence at page 16, line 293:

*This parameterization can easily be incorporated in a DFN modeling framework to investigate the hydraulic responses at reservoir scales, assuming that the minimal correlation length is no longer than $1/16$ of the reservoir size. If DFN's of scales close to the correlation length are considered, fluctuations of the flow behaviour have to be taken into account (e.g., de Dreuzy et al., 2012), as this can modify network flow connectivity.*

**Various comments on other points along the text:**

\* The introduction is rather short, but logically organized, and provides a proper summary of the state of the research on the topic so far. The authors use an approach relying on a large statistics of fracture with identical. They could mention that the first approach of this kind was proposed by Méheust and Schmittbuhl in a JGR paper in 2001, studying populations of synthetic rough fractures with self-affine aperture fields (that is, for lc/L=1).

We changed page 3, lines 59 – 63 to:
*Following Méheust and Schmittbuhl (2001,2003) the ratio between system size L, and the correlation length lc defines whether the fracture has an intrinsic permeability or not. Their statistical approach suggested that permeabilities of uncorrelated fractures (i.e., lc/L = 1) are strongly fluctuating and anisotropic for the same roughness configurations, revealing the importance of considering low lc/L ratios to be able to quantify an intrinsic fracture permeability.*

\* In the presentation of Eq. (1), the hypothesis of permanent flow is missing.

[Figure]

We changed page 3, line 76 to:

*..., i.e., momentum balance (1) and continuity (2) equations, which for steady-state flow conditions are given in compact form by: ...*

\* In Eq. (6), the mathematical notation is strange: a is used both for the aperture field prior to negative values being put to 0, and for the the aperture field whose negative values have been put to 0. Of course when coding one may use the same variable name, overwriting the previous variable a, but mathematically they are two different quantities.

Of course, thanks for pointing out.
We changed the notation from $a(x, y)$ to $a_0(x, y)$ on page 4, lines 94 and 95 (eq. 6), page 5, lines 110 (eq. 9) and page 5, line 122.

\* Page 5, line 104: "leaving $H$ as a measure for the intensity of small scale roughness".

This explanation is a bit caricatural. $H$ is rather a measure of the ratio between larger scale roughness and smaller scale roughness (the ratio being always larger than 1 since h>0, but all the smaller as H is smaller).

You are right, this might sound over simplified.
We changed page 5, line 104 to:

*..., leaving H as a measure for the ratio of large scale versus small scale roughness intensity.*

\* Equation (8): some authors choose to divide the standard deviation of the aperture by the mechanical aperture, which is the mean aperture prior to putting negative

apertures to 0 and thus corresponds to the distance between the mean planes of the facing topographies. Is there a particular reason why you chose to use the mean aperture?

The reason for that is, that we wanted to have a parameter, that could be computed from in-situ fracture data (e.g. CT-scans). There, it is very difficult to acquire information of the individual fracture surfaces, as one can only compute it from the entire void space. Also, we think that it better reflects effective properties of a fracture, as for fractures with large closures there might be trapped pore space within the fracture (we shortly addressed this on page 8, line 160).

* Page 4, line 113: why don't you express the condition of contact in terms of R $(R \geq 1/(3/\sqrt{2})$?

Thanks for pointing it out. In that context, it is better to express it in terms of R, rather than mean aperture.
We changed page 5 line 113 to:
. . . *and the surfaces are in contact if* $R \geq (3\sqrt{2})^{-1}$ *(see Brown, 1987).*

* Page 5, line 121: It seems that a simple way of presenting S would be as the ratio of the fracture surface's area to twice that of its projection on the fracture plane.

Yes, exactly. This is what is shown with the term $\frac{sa_f}{sa_c}$ in equation 10. However, you still have to normalize by the contact fraction.

We simplified the text on page 5 line 121 according to your suggestion:
*For that, we calculate the ratio of the surface area of the fracture* $sa_f$ *to twice the area*
*of its projection on the fracture plane (i.e., two times the base area perpendicular to the flow direction) $sa_c$ and normalize it with the fractional amount of the aperture field that has opened, i.e. ...*

\* Page 6: was the conservation of the total volumetric flow rate tested ? What are the relative flow rate fluctuations between all sections transverse to the mean flow ?

We understand the concern of the reviewer, that the numerical solution procedure might introduce errors in the mass conservation. In this case the result would be fluctuations in the relative flow-rate transverse to the principal flow direction. While computing the divergence of the continuity equation (eq. 2), we make sure, that the absolute mass-conservation residual is reduced down to $10^{-8}$. Visualizing the spatial distribution of the mass-conservation residual (see figure 3 for an example) demonstrates this. Additionally, in Appendix A of Eichheimer et al., 2019 (https://doi.org/10.5194/se-10-1717-2019) it has been demonstrated that relative residual reductions lower than $10^{-7}$ deliver constant permeability values (using the same numerical code). Based on this, we conclude that the obtained solutions are sufficiently correct.

\* Table 1: It would be interesting to have the mininum and maximum values of R in the table.

See comment above, we inserted a new table on page 9 containing that information.

\* Page 8, line 184-185, about the inset plot: the contact fraction is only controlled by the PDF of apertures prior to setting negative values to 0; that PDF is mostly independent of lc/L (though if one looks closely one may find a slight dependence),

and therefore only dependent on the fracture closure. This is well known.

We agree. Because of that, we shifted the percolation analysis to the description of the synthetic dataset, rather than presenting it as new results (see comment above). However, we now know that this also holds for the way we compute the mean aperture (setting negative values to zero and then computing the PDF).

* Page 12, line 225: Here you probably mean "perpendicular to the fracture plane", i.e., the vertical direction if the fracture is horizontal.

In that case, the result of both expressions is the same. By applying pressure boundary conditions on two opposing fracture walls while setting the remaining to no-slip, we enforce a principal flow direction which in that case is parallel to the fracture plane. For porous media, however, this is not the case which is why we decided to keep the expression as is, but refining page 12, line 225 by:
*. . ., the resolution perpendicular to the principal flow direction is the most crucial part.*

* Equation (8): in this equation, it seems that the norm is simply the absolute value of the relative error. Why use a square inside a root mean square ?

Yes, you're right – thanks for pointing out.
We changed the notation of equation 18 to:
$||\delta_k|| = |\frac{k_r - k_{max}}{k_{max}}|$

* Discussion of page 14: here you could mention that the lower the value of $l_c/L$, the larger the impact of the vertical flow tortuosity on the fracture's permeability.
We changed page 14, lines 251-253 to:

*Considering flow predictions for uncorrelated fractures (i.e. $l_c/L \geq 1$) is problematic. Blocked pathways connected to the early appearance of the percolation limit (see Fig. 4) or flow enhancing configurations ($\chi > 1$) as also observed by Méheust and Schmittbuhl (2000) are producing substantial variations in their hydraulic efficiencies. With decreasing lc/L ratios, the impact of vertical flow tortuosity on its permeability increases due to a larger portion of flow inhibiting configurations compared to flow enhancing ones (see Méheust and Schmittbuhl, 2000). On the contrary, the fluctuations in the average flow behavior decrease significantly with decreasing lc/L ratios.*

\* Page 15, line 262: "correlation lengths that are equal to the size of the fracture .... seem rather unrealistic".

The origin of the correlation length is not generally known, is it ? Is it mechanical ? A fresh fracture without shift along the fracture plane would present a constant aperture field, one with a shift of length l would have a correlation length lc = l in that direction, but then the aperture field would be anisotropic.

We agree that little is known about the origin of the correlation length, but we follow the hypothesis of Brown, 1995, that it originates from mechanical principles. We simply wanted to make the point that a full fracture with a correlation length equal to the size of the fracture is unrealistic. If we considering the study of Schultz et al., 2008 (DOI: https://doi.org/10.1016/j.jsg.2008.08.001), the displacement of joints ($d$) scales with its length ($L$) by: $d = \alpha L^{0.5}$ with $\alpha$-values in the order of 0.01 to 0.001. Assuming that the correlation length scales with the fractures displacement, results in maximal correlation-length-to-fracture-size ratios of 0.01. So it is not possible to have fractures

with correlation length that is equal to their size. This highlights the need for further research on that topic.

To elaborate this in the paper, we changed page 14, lines 261-264 to:

*From a mechanical perspective, correlation lengths that are equal to the size of the fracture seem rather unrealistic, considering that the shear displacement $ds$ of fractures scales with their length $Lf$ according to $ds = \alpha Lf^{0.5}$ (Schultz et al., 2008). Using $\alpha$ values between 0.01 and 0.001, which is about the range for Moros joints in Schultz et al. (2008), results in a maximal lc/Lf ratio of ratio of 0.01. This thus illustrates that further research on fractures correlation lengths is required because the presence of it is omnipresent in most relevant studies ...*

\* Page 15, line 269: Here and elsewhere I would use "parallel plate equivalent" (which refers to the geometry) rather than "cubic law equivalent", which involves a hydraulic concept. The two fractures are equivalent in that their mean apertures are identical (a geometric feature), not in that their hydraulic behavior is the same (this equality defines the fracture's hydraulic aperture).

Indeed, using the term "cubic law equivalent" could be misleading. We changed it throughout the paper to your suggestion at page 5, line 113 and line 119, page 6, line 123, page 9, line 195 and 196 and page 15, line 269.

**Writing:**

\* Shouldn't the vectorial quantities (including nabla) appear in bold fonts?

We are not sure about that and would leave the final decision to the editor.

\* Page 4, line 100: I would call the "rescaling factor" simply a "prefactor".

We incorporated the suggested change at page 4, line 100.

\* Page 6, line 137: ")" should be removed after "0.01 Pa".

This was already changed in RC1.

\* Page 6, line 145: here I'd write "with $\eta$ the fluid's dynamics viscosity".

We incorporated the suggested change at page 6, line 145. We changed the symbol for viscosity to $\mu$ to be consistent (This was already mentioned in the RC1).

\* Page 8, line 176: I think "build" should be "built" here; please check.

Yes, correct. We corrected it at page 8, line 176.

\* Page 12, line 219: "multiplied by" rather than "on".

We incorporated the suggested correction at page 12, line 219.

\* Page 12, line 225: "the resolution perpendicular to the flow direction".

See comment above, we changed Page 12, line 225 to:

*. . ., the resolution perpendicular to the principal flow direction is the most crucial part.*

[Figure]

**Fig. 1.** Mean of the hydraulic efficiency for different correlation-length-to-size ratios

[Figure]

**Fig. 2.** Standard deviation of the hydraulic efficiency for different correlation-length-to-size ratios

[Figure]

**Fig. 3.** Spatial distribution of the residual of the continuity equation (eq. 2) after the numerical solution process

---

## Editor Comment (EC1) · Randolph Williams (Editor) · 27 Apr 2020

Please see these additional comments from reviewer #2 on the revised version of this manuscript, which were sent to me via email.

* Line 299: this parametErization" –> spurious E.

* Lines 300-301: I have the feeling that the message "fluctuations of the flow behavior have to be taken into account" is not exactly proper. If Lc is on the order of some scales in the DFN then the DFN's permeability can simply not be computed as the product of

the permeability of the corresponding parallel plate DFN (where each fracture is re-placed by the parallel plate of identical mean aperture) and a factor accounting for the mean permeability reduction due to roughness. Precisely due to changes in net-work SCALE flow connectivity (in you sentence on line 302 "scale" is missing between "network" and "flow").

* Line 224: my comment was not about the boundary conditions but more about the fact that the smallest dimension is along the fracture aperture and therefore it is along that direction that the highest shear occurs. So it is the discretization across the aperture that matters most when solving numerically.

* Lines 251-253: it is also because at small Lc/L the in-plane tortuosity acts at a scale that is not much larger than the scale at which the vertical tortuosity acts. In-plane flow channeling has a much reduced impact on transmissivity, and thus vertical flow tortuosity has relatively more impact.

---

## Author Comment (AC3) · 28 Apr 2020

We sincerely thank the referee for reviewing the revised manuscript. Please find below a point-by-point response to the referee comments (comments of the reviewer in black and our response in red, text changes appear in italic font)

On behalf of all authors, yours sincerely,
Maximilian Oskar Kottwitz

[Figure]

\* Line 299: this parametErization" – spurious E.

Thanks for pointing out, we removed an "e" at the indicated location

\* Lines 300-301: I have the feeling that the message "fluctuations of the flow behavior have to be taken into account" is not exactly proper. If Lc is on the order of some scales in the DFN then the DFN's permeability can simply not be computed as the product of the permeability of the corresponding parallel plate DFN (where each fracture is replaced by the parallel plate of identical mean aperture) and a factor accounting for the mean permeability reduction due to roughness. Precisely due to changes in network SCALE flow connectivity (in you sentence on line 302 "scale" is missing between "network" and "flow").

We revised the sentence at lines 300-302 to:
*If DFN's of scales close to the correlation length are considered, fluctuations in the average flow behaviour are expected. This can modify network scale flow connectivity and thus requires additional concepts to compute permeabilities (e.g., de Dreuzy et al., 2012)*

\* Line 224: my comment was not about the boundary conditions but more about the fact that the smallest dimension is along the fracture aperture and therefore it is along that direction that the highest shear occurs. So it is the discretization across the aperture that matters most when solving numerically.

Yes you're right. We made the sentence at line 224 clearer according to your suggestion:
*For numerical permeability estimations of single fractures, the resolution perpendicular to the aperture field is the most crucial part.*

\* Lines 251-253: it is also because at small Lc/L the in-plane tortuosity acts at a scale that is not much larger than the scale at which the vertical tortuosity acts. In-plane flow channeling has a much reduced impact on transmissivity, and thus vertical flow tortuosity has relatively more impact.

We incorporated your suggestion at line 253:
*With decreasing $l_c/L$ ratios, the impact of vertical flow tortuosity on its permeability increases relative to the impact of in-plane tortuosity, as both start to act at comparable scales and generally the fractures exhibit larger portions of flow inhibiting regions compared to flow enhancing ones (see Méheust and Schmittbuhl, 2000).*